# Musculoskeletal Injury Risk in a Military Cadet Population Participating in an Injury-Prevention Program

**DOI:** 10.3390/medicina59020356

**Published:** 2023-02-13

**Authors:** Ivar Vähi, Leho Rips, Ahti Varblane, Mati Pääsuke

**Affiliations:** 1Estonian Military Academy, War and Disaster Medicine Centre, Riia 12, 51010 Tartu, Estonia; 2Institute of Sport Sciences and Physiotherapy, Faculty of Medicine, University of Tartu, Ujula 4, 51008 Tartu, Estonia; 3Sports Medicine and Rehabilitation Clinic, Tartu University Hospital, Puusepa 1a, 50406 Tartu, Estonia; 4Institute of Clinical Medicine, Faculty of Medicine, University of Tartu, L. Puusepa 8, 50406 Tartu, Estonia

**Keywords:** injury prevention, warm-up, exercise, military training

## Abstract

*Background and Objectives:* Musculoskeletal injuries are a major health hazard among military personnel. Previous research has proposed several exercise-based strategies for prevention. The purpose of this study was to investigate the effect of an exercise-based injury-prevention program on the incidence of musculoskeletal injury, motor performance and psychosocial status. *Materials and Methods:* Thirty-six Estonian Military Academy cadets were randomly assigned into either an intervention or control group. The intervention group followed a neuromuscular exercise-based injury-prevention warm-up program, three times per week for 6 months. The control group continued with the usual warm-up. The main outcome measure was injury incidence during the study period. Additionally, evaluation of isokinetic lower-extremity strength, postural sway, physical fitness and psychosocial status was included pre- and post-intervention. *Results:* During the 6-month study period, the musculoskeletal injury incidence was 43% in the intervention group and 54% in the control group (RR = 0.8; 95% CI = 0.41 to 1.99). The noted 20% risk reduction was not statistically significant (*p* = 0.59). Furthermore, there were no statistically significant differences between the intervention and control group in motor performance or psychosocial status measures. *Conclusions:* In conclusion, no effect of the exercise-based injury-prevention program on injury risk, motor performance or psychosocial status could be detected.

## 1. Introduction

Musculoskeletal injuries (MSI) are a common health problem among military personnel [1,2,3,4]. In 2019, 55% of the United States Army active component soldier population reported an injury, and 72% of these were considered to be overuse injuries [2]. Injury incidence varies between different unit types and can be as high as 65.6% [5]. Jones et al. [4] reported injuries to be the leading cause of medical encounters and the most frequent medical problem affecting service members. MSIs result in a high rate of limited active-duty days [2,3], raised medical cost [5] and attrition due to medical reasons [6].

MSI risk factors among military personnel include, but are not limited to, exercise- and sport-related activities [3], faulty biomechanical movement patterns [7] and low physical fitness [8]. It is necessary to understand that physical activity, which is necessary to improve physical fitness, is itself one leading risk factor for MSI [9]. Therefore, finding safe means to develop physical fitness should be a priority.

Research has proposed several strategies for MSI prevention. For example, bracing [10], mouthguards [11], rule modifications [12], dietary calcium and vitamin D supplementation [13], exercise-based interventions [14,15,16] and decreased running frequency, duration and distance [16,17] have previously resulted in decreased injury incidence. The rationale for implementing exercise-based injury-prevention programs is to target modifiable intrinsic risk factors such as possible deficits in the biomechanics and physical fitness of individuals [14,15,16,18]. Exercise-based injury-prevention intervention could be used as a structured training program as a whole [16], a comprehensive warm-up strategy before physical activity [19] or an addition to usual training [14,15,20].

Among athlete and youth populations, research shows that exercise-based programs protect against injury [21,22]. Results in the military population, however, seem to be somewhat mixed regarding the effect of exercise programs on MSI incidence. Several studies have found exercise interventions to reduce injury incidence [14,15,16,23,24], while others have reported that intervention has no statistically significant effect on injury incidence [25,26,27,28,29]. For example, physical-readiness training reported by Knapik et al. [16] resulted in reduced injury risk. Neuromuscular training in combination with supervised gait retraining and biofeedback has shown a reduced risk of medial tibial stress syndrome [14]. Selected strength exercises combined with static stretching, in addition to formal physical training sessions, showed a reduced risk of anterior knee pain in British Army recruits [23]. Parkkari et al. [15] reported that neuromuscular training in combination with injury-prevention education reduced acute ankle and upper-extremity injuries significantly. However, contrary to these studies resulting in reduced injury risk, Carow et al. [25] did not find any significant effect of neuromuscular exercise-based warm-up on injury risk. Goodall et al. [27] reported that intervention consisting of balance and agility exercises did not have a statistically significant effect on lower limb, knee and ankle or knee and ankle ligament injury incidence. Childs et al. [30] reported no significant effect of a core-stabilization exercise program on injury risk. According to Dijksma et al. [31], adding stretching to training has a small and nonsignificant effect on total injury risk. However, Amako et al. [24] reported that stretching reduced muscle/tendon and overuse injury risk. Dijksma et al. [31] also pointed out that the current evidence regarding exercise-based injury-prevention program effectiveness in military populations is of low quality. Brushøj et al. [29] were not able to detect any effect of an injury-prevention program containing strength, flexibility and coordination exercises on overuse knee injury and medial tibial stress syndrome incidence. However, interestingly, soldiers participating in a prevention program significantly improved their running distance in a 12-min test [29]. It seems that programs that may be effective in reducing injury risk in military populations are multifactorial in nature—neuromuscular training programs with a combination of mobility, stretching, strength, agility, balance and plyometric exercises [14,15,16,23].

There seems to be inconsistency in previous research regarding MSI prevention in the military population. Furthermore, there is a lack of studies covering MSI prevention via exercise-based injury-prevention programs in the Estonian Military. Finally, MSI is a major health hazard in the military population. Therefore, this randomized, prospective study was designed. The primary purpose of this study was to investigate the effect of an exercise-based injury-prevention program on MSI incidence. The secondary purpose was to investigate whether the prevention program had any influence on motor performance or psychosocial status.

## 2. Methods

### 2.1. Study Design

This study was a randomized, prospective study assessing the effect of an exercise-based injury-prevention program on MSI incidence, motor performance and psychosocial status. The recruitment period was 15 December 2019 to 1 March 2020. The study period was 6 months (2 March 2020 to 16 August 2020). Outcome variables were measured and recorded at the baseline and after the 6-month study period. This study received ethical approval from the University of Tartu Research Ethics Committee (No. 298/T-10, 18.11.2019). This study was registered at clinicaltrials.gov (Identifier: NCT05684003).

### 2.2. Participants

The participants in this study were Estonian Military Academy cadets. Potential participants were briefed on the study objectives. All participants provided written informed consent prior to enrollment. Participants were included in the study if they had no current injuries causing limited duty days or inability to participate in the proposed injury-prevention program. Participant health status, in accordance with set criteria, was assessed by an orthopedic surgeon. All participants were randomly assigned to one of two groups: an intervention (INT) group that followed the injury-prevention program and a control (CON) group that continued their usual physical training without the additional injury-prevention program. The participant flow through the study is presented in Figure 1. Since there were only 3 female participants and 6 male participants were unable to attend laboratory motor performance assessment, their data were omitted and the final sample for analysis consisted of 14 male participants in the INT group and 13 male participants in the CON group. Participant baseline anthropometric characteristics are presented in Table 1.

### 2.3. Injury Incidence

The primary outcome of interest in our study was incidence of musculoskeletal injury during the 6-month study period. Injury tracking was performed through the Estonian Defence Forces medical database. Since participants receive health care through the military health care system, any musculoskeletal condition severe enough to seek treatment from a medical provider is registered. Additionally, participants were asked to recall any musculoskeletal conditions at 3 and 6 months from the beginning of the study in order to catch potential injuries not reported to a military medical provider, or for which they might have obtained treatment from a nonmilitary medical care provider.

### 2.4. Motor Performance Testing

Prior to motor performance testing, baseline body mass (kg) and height (cm) were measured. Based on these, body mass index (BMI) was calculated (BMI = weight (kg)/height (m)^2^). Before testing motor performance participants performed a 10-min individual warm-up—jogging, and a range of motion and bodyweight strength exercises.

### 2.5. Isokinetic Muscle Strength Testing

Strength parameters of the knee flexor and extensor muscles were recorded using a computerized dynamometer (CON-TREX^®^ MJ, Physiomed Elektromedizin AG, Schnaittach, Germany). Participants were seated upright with a hip flexion of 85°, the trunk stabilized with two 3-point seat belts and arms folded across the chest. The extremity to be tested was supported at the distal thigh with a Velcro strap. The distal shin adapter was attached 2–3 cm proximal to the lateral malleolus using a Velcro strap. The alignment of the rotational axis of the dynamometer was set so it would pass transversely through the femoral condyles, 1.5 finger width above joint space and vertical above head of fibula. The peak torque of knee flexion and extension was measured in concentric mode at an angular velocity of 60°/s between a knee flexion of 10° and 90° (0° = full extension). For task-specific warm-up, the participants performed three submaximal knee flexion–extension movements. The test procedure included 3 maximal, consecutive flexion–extension movements. The participants were given verbal encouragement to flex and extend the leg as fast and forcefully as possible. Based on flexion–extension peak torque measurements, the hamstring/quadriceps ratio was calculated (H:Q = (flexion peak torque/extension peak torque) × 100%). Additionally, H:Q and peak torque differences between left and right lower extremity were calculated.

### 2.6. Postural Sway Measurement

A force platform (Kistler 9286A, Switzerland, dimensions 40 × 60 cm) and biomechanical movement analysis system Elite Clinic with Sway software^®^ (BTS S.p.A., Milan, Italy) were used to assess the equivalent area (EqArea) of the center of pressure sway. Participants were asked to maintain balance while standing on a single leg for 30 s. For familiarization, the participants balanced 30 s on each leg. Based on the measurements obtained, the difference of the EqArea between the left and right legs was calculated.

### 2.7. Physical Fitness Test

Participant physical fitness was assessed with a test consisting of a 2-min maximal-effort push-up event, a 2-min maximal-effort sit-up event and a 3.2 km timed run.

### 2.8. Psychosocial Status

To assess participant psychosocial status, the RAND 36-item Health Survey 1.0 was used. All items in this survey are scored on a scale from 0 to 100, with a high score defining a more favorable health state. For analysis, the following subscales were used: physical functioning, role limitations due to physical health problems, role limitations due to personal or emotional problems, energy/fatigue, emotional well-being, social functioning, bodily pain and general health perception.

### 2.9. Exercise Intervention

Participants in the INT group were asked to follow a prescribed injury-prevention program (PP) during warm-up before physical training for 6 months, 4 times per week. The PP was a structured warm-up routine, taking 20 min to finish. It was divided into 3 sequential parts: (1) whole-body range of motion and bodyweight strength exercises, (2) aerobic load—jogging, and (3) jump, balance and running exercises with a change of direction. A written PP description is provided in Appendix A. INT group participants were instructed on correct PP execution during a workshop led by a physical therapist. Furthermore, detailed written instructions and online video materials were provided to ensure correct program execution. During the study period, no supervision on PP execution technique was provided. Participants in the CON group were asked to continue with their regular training and warm-up without any restrictions. The usual warm-up in the CON group involved a combination of aerobic exercises and range of motion exercises.

### 2.10. Statistical Analysis

Descriptive statistics (mean, SD) in both groups were calculated. Differences between INT and CON groups were analyzed using an independent sample Student’s *t*-test, and within-group differences between time points were analyzed using a dependent-sample paired *t*-test if the assumptions of a normal distribution and homogeneity of variances were satisfied. Normality of distribution was controlled using the Shapiro–Wilk test. Homogeneity of variances was controlled with Levene’s test. If the previous assumptions for the *t*-test were not met, the between-group differences were analyzed with the non-parametric Wilcoxon test, and within-group differences between time points were analyzed using the dependent-sample paired Wilcoxon test. Bonferroni correction was applied to account for possible false positives due to the number of tests. Relative risk (RR) and 95% confidence intervals (95% CI) for between-group injury incidence were calculated. The level of significance was set at *p* < 0.05. R version 4.1.1 and RStudio version 1.4.1717 were used for statistical analysis.

## 3. Results

There were no significant group differences between the INT and CON groups at baseline. During the 6-month study period, the musculoskeletal injury incidence was lower in the INT group compared with the CON group. The noted 20% risk reduction was not statistically significant (*p* = 0.59). The most frequent injury type (acute vs. overuse) in both the INT and CON groups was overuse injuries, comprising 77% and 94% of injuries, respectively. The difference between these proportions was not statistically significant (*p* = 0.85). The most frequent injury location (lower extremity vs. back vs. upper extremity) in both the INT and CON groups was lower extremity, comprising 85% and 50% of injuries, respectively. The difference between these proportions was not statistically significant (*p* = 0.6). Detailed injury data is presented in Table 2 and Table 3. The INT group compliance rate was 77.5%, meaning that, on average, the PP was used 3.1 times per week. There were no statistically significant differences between groups in motor performance or psychosocial status measures. Detailed data concerning these last two measures are presented in Table 4.

## 4. Discussion

Based on the results in our study, we were unable to detect any effect of an exercise-based injury-prevention program on MSI incidence in the military cadet population. One of the reasons why significant differences in injury incidence were not detected might be related to the prevention program specifics. This study, in contrast to some previous studies, did not have any progression (e.g., increase in number of repetitions) built into the program [19,23]. Furthermore, in our study, the intervention was designed as a warm-up, whereas, for example, Knapik et al. [16] designed their program as a complete standardized neuromuscular training program. Consequently, their intervention also controlled for external injury risk factors, namely training load and its progression. Furthermore, based on the work of Carow et al. [25] and Coppack et al. [23], expert supervision in the execution of the prevention program might have contributed to reduced injury incidence. Even when we grouped injuries by location or type, no significant differences in injury incidence were seen. Since our intervention was designed to be a comprehensive warm-up routine for the entire body, it might have lacked the specificity and overload to have a significant effect on more specific injury types or locations. There are various risk factors for MSI [7,8]. Their effect on injury incidence may come also through more complex interactions, and this effect can be different at the individual level [32]. It might be that our program, which was intended to correct possible faulty biomechanical movement patterns and improve physical fitness, lacked a preventive effect because our participants did not have significant negative deviations in those areas; this means that a more specific participant selection to filter out those with deficits in motor performance, and thus at bigger risk of sustaining MSI, could yield different results.

An additional purpose of the study was to investigate whether the prevention program had any influence on motor performance or psychosocial status of study participants. Motor performance assessment was included because our MSI-prevention program was designed to correct potential deficits in biomechanics and physical fitness. Psychosocial assessment was included because it has been reported that emotional problems and stress contribute to MSI risk and should be considered when planning preventive measures [33,34]. We hypothesized that our program might increase confidence through better movement control, and decrease potential musculoskeletal discomfort, which in turn could improve emotional and social well-being. In this study we did not find any effect of the exercise-based injury-prevention program on psychosocial status. Previous research has shown that implementing psychological interventions to reduce stress and anxiety reduces injury incidence [34,35]. Based on this, one should consider designing an injury-prevention program that has both psychological and physical components in order to address potential deficits in biomechanics and also possible mental problems. Contrary to our study, previous research has shown that neuromuscular exercise-based injury-prevention warm-up programs result in improved motor performance, e.g., proprioception [36], balance [37], strength [37], jumping speed [38], vertical jump [39], 2-mile run time [40], Army Physical Fitness Test push-up and total score [40] and agility [39]. This supports the concept of reducing MSI with exercise-based prevention programs aimed at correcting possible deficits in biomechanics and physical fitness. Additionally, enhanced physical fitness would most likely further motivate participation in such a program. When comparing program specifics of this study and those previously resulting in improved physical fitness, the main differences seem to be that these programs had a higher plyometric load [37,38,40], a higher resistance-training load [36,37], program execution with supervision [40], no mobility exercises [36,37,40], exercise progression with increasing level of difficulty [23,36,38] and the use of training equipment [38]. Overall, it seems that our program might have lacked the necessary load required for training adaptations to occur. We did not use training equipment because we wanted the prevention program to be ready to use regardless of the availability of such means. Furthermore, having no supervision was intentional because we believed that eventually the program, which could be followed after initial instruction individually, would have fewer barriers for implementation.

The strength of our study is that we assessed the effect of an exercise-based injury-prevention program simultaneously on injury incidence, motor performance and psychosocial status. Previous research has focused on just a single outcome area, making it somewhat difficult to understand which element has resulted in a reduced injury risk. Furthermore, the 6-month study period should be considered a strength of the current study. This study also has several limitations. Although not an exclusion criterion, due to the small female sample size, female participants were not included in the analysis, which prevents generalizability of our findings to females. Furthermore, the statistical power of this study was low due to the small sample size.

## 5. Conclusions

Based on the data presented in this study, we conclude that no effect of a neuromuscular exercise-based injury-prevention warm-up program on injury risk, motor performance and psychosocial status could be detected. In this study, our primary focus was exploring the effect of an exercise-based MSI-prevention program on injury incidence. In future research, the effect of combining psychological and exercise components in MSI-prevention programs should be examined.

## Figures and Tables

**Figure 1 medicina-59-00356-f001:**
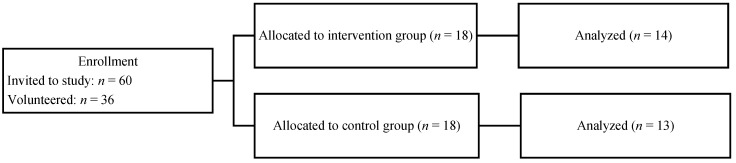
Flow of participants through the study.

**Table 1 medicina-59-00356-t001:** Baseline anthropometric characteristics.

	INT (*n* = 14)	CON (*n* = 13)
Age (years)	23.4 ± 3.2	23.1 ± 4
Body mass (kg)	84.6 ± 8.6	83.4 ± 9.7
Height (cm)	182.9 ± 5.6	181.4 ± 4.9
Body mass index	25.3 ± 2.5	25.5 ± 3.8

Data are mean ± SD; INT, intervention group; CON, control group.

**Table 2 medicina-59-00356-t002:** Injury incidence.

	INT Group (*n* = 14)		CON Group (*n* = 13)				
Injury Outcome	Injured—no.	Proportion—%	Injured—no.	Proportion—%	RR	95% CI	*p* Value
All injuries	6	43%	7	54%	0.8	0.41; 1.99	0.59
Acute	3	21%	1	8%	2.79	0.18; 13.18	0.34
Overuse	6	43%	7	54%	0.8	0.41; 1.99	0.59
Lower extremity	6	43%	5	38%	1.11	0.42; 2.62	0.84
Back	1	7%	2	15%	0.46	0.07; 7	0.53
Upper extremity	1	7%	2	15%	0.46	0.07; 7	0.53

INT, intervention; CON, control; RR, relative risk; CI, confidence interval.

**Table 3 medicina-59-00356-t003:** Number of injuries.

	INT Group		CON Group		
	Injuries—no.	Proportion—%	Injuries—no.	Proportion—%	*p* Value
All injuries	26	100%	18	100%	0.92
Acute	6	23%	1	6%	0.31
Overuse	20	77%	17	94%	0.85
Lower extremity	22	85%	9	50%	0.6
Back	2	8%	7	39%	0.53
Upper extremity	2	8%	2	11%	0.59

INT, intervention; CON, control.

**Table 4 medicina-59-00356-t004:** Motor performance and psychosocial status.

	INT (*n* = 14)			CON (*n* = 13)			6 Months		
	Baseline	6 Months	*p*	Baseline	6 Months	*p*	INT (*n* = 14)	CON (*n* = 13)	*p*
**Motor performance**									
**Postural sway**									
Right EqArea (mm^2^)	9178.9 ± 3712.4	8203.7 ± 2744.2	0.3	9761.8 ± 3459	7090.6 ± 1606	0.005	8203.7 ± 2744.2	7090.6 ± 1606	0.38
Left EqArea (mm^2^)	8364.6 ± 2750.3	8771.4 ± 1832.1	0.54	7665.5 ± 2906.2	8212.2 ± 2086.2	0.19	8771.4 ± 1832.1	8212.2 ± 2086.2	0.22
Difference in EqArea (mm^2^)	2080.6 ± 2386.2	2097.6 ± 1304.9	0.58	3382.2 ± 3138.8	2234.8 ± 1919.2	0.33	2097.6 ± 1304.9	2234.8 ± 1919.2	0.98
**Isokinetic strength**									
PTE left (Nm)	217.9 ± 35.9	200 ± 47.6	0.07	223.3 ± 46.5	204.1 ± 25.9	0.1	200 ± 47.6	204.1 ± 25.9	0.65
PTF left (Nm)	130.5 ± 22.1	122.1 ± 28.9	0.18	135.6 ± 31.2	129.1 ± 19.4	0.64	122.1 ± 28.8	129.1 ± 19.4	0.4
left H/Q ratio (%)	60.6 ± 10.2	61.7 ± 9.2	0.72	61.3 ± 10.7	63.5 ± 7.6	0.27	61.7 ± 9.2	63.5 ± 7.6	0.62
PTE right (Nm)	207.5 ± 47.3	204.6 ± 54.8	0.7	231 ± 54.4	210 ± 24.1	0.07	204.6 ± 54.8	210 ± 24.1	0.79
PTF right (Nm)	131.5 ± 31	135.4 ± 35.7	0.42	144.4 ± 28.8	140.9 ± 26.6	0.48	135.4 ± 35.7	140.9 ± 26.6	0.68
right H/Q ratio (%)	64.9 ± 15.1	67.2 ± 10.5	0.36	63.3 ± 7.9	66.9 ± 8.1	0.21	67.2 ± 10.5	66.9 ± 8.1	0.65
PTE difference (Nm)	26.9 ± 18	28.8 ± 17.5	0.76	16.6 ± 15.4	21.3 ± 15	0.27	28.8 ± 17.5	21.3 ± 15	0.19
PTF difference (Nm)	16.2 ± 18.8	15.9 ± 14.8	0.9	18.6 ± 14.2	20.5 ± 20.3	0.8	15.9 ± 14.8	20.5 ± 20.3	0.73
H/Q ratio difference (%)	11.4 ± 11.9	6.1 ± 4.7	0.5	12.1 ± 8.2	7.5 ± 5.7	0.14	6.1 ± 4.7	7.5 ± 5.7	0.76
**Physical fitness test**									
Push-up (repetitions)	66.2 ± 7.1	65.9 ± 12.3	0.93	67.8 ± 8.8	65.9 ± 11.1	0.67	65.9 ± 12.3	65.9 ± 11.1	0.98
Sit-up (repetitions)	63.2 ± 14	68 ± 13.7	0.05	69.2 ± 12.4	65.7 ± 14.2	0.29	68 ± 13.7	65.7 ± 14.2	0.87
3.2 km run (min)	14.4 ± 0.9	14.6 ± 0.9	0.5	14.4 ± 1.7	15.4 ± 2.2	0.03	14.6 ± 0.9	15.4 ± 2.2	0.27
**Psychosocial status**									
**RAND 36-item survey scores**									
physical functioning	96.1 ± 4.5	97.9 ± 4.7	0.43	94.2 ± 9.5	93.9 ± 8.2	0.89	97.9 ± 4.7	93.9 ± 8.2	0.1
role limitations due tophysical health problems	76.8 ± 31.7	91.1 ± 27.1	0.11	78.9 ± 32	90.4 ± 28	0.26	91.1 ± 27.1	90.4 ± 28	0.97
role limitations due toemotional problems	59.5 ± 45.6	81 ± 38.6	0.11	92.3 ± 20	100 ± 0	0.37	81 ± 38.6	100 ± 0	0.09
energy/fatigue	56.4 ± 13.2	62.9 ± 12.4	0.12	53.9 ± 13.9	60 ± 13.5	0.1	62.9 ± 12.4	60 ± 13.5	0.45
emotional well-being	78 ± 12.9	82.9 ± 9.2	0.14	79.4 ± 10.7	80.6 ± 9.8	0.74	82.9 ± 9.2	80.6 ± 9.8	0.56
social functioning	85.7 ± 26.8	84.8 ± 22	1	90.4 ± 14.6	88.5 ± 14.8	0.77	84.8 ± 22	88.5 ± 14.8	0.87
bodily pain	76.3 ± 12.1	85 ± 16.4	0.03	76.5 ± 24.2	86.5 ± 15.7	0.09	85 ± 16.4	86.5 ± 15.7	0.8
general health	75.4 ± 8.2	79.6 ± 9.3	0.22	62.3 ± 17.6	68.1 ± 17.5	0.1	79.6 ± 9.3	68.1 ± 17.5	0.08

Data are mean ± SD. EqArea, equivalent area; PTE, peak torque in extension; PTF, peak torque in flexion; H/Q, hamstring to quadriceps. After Bonferroni correction, the necessary value for a significant difference was *p* < 0.001.

## Data Availability

The datasets used and/or analyzed during the current study are available from the corresponding author on reasonable request.

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
