# Peer review of "Musculoskeletal Injury Risk in a Military Cadet Population Participating in an Injury-Prevention Program"

_medicina, 2023, doi:10.3390/medicina59020356_

Round 1

Reviewer 1 Report

Thank you for the opportunity to review your research. Excellent methodology, relevant and precise study.

The research is impressive, especially in its construction and implementation, including carefully monitoring all parameters and the statistical analysis.

The only concern is about the sample size, which on the face of it, is small. The sample size can affect the study's results and conclusions.

English language and style are great.

Author Response

Dear reviewer,

We appreciate your time in handling our manuscript and thank you for the comments. We agree that sample size should be taken into consideration while interpreting the results of this study. To guide the reader, this issue is pointed out as one of the limitations of this study. At the same time we still believe that the results of this study are adequate and of value.

Reviewer 2 Report

Thank you for your article. I only have minor grammar corrections. In my opinion you have executed the study in the correct way, I would have chosen a different exercise program and many more participants. The value of your study is that it shows what doesn't work.

Author Response

Dear reviewer,

We appreciate your time in handling our manuscript and thank you for the comments. We revised our manuscript according to your suggestions regarding grammar except your first suggestion.

You suggested to add „in the military“.

Our response: no correction was made because this previous research that we are referring there is not exclusively military based.

We agree that sample size should be taken into consideration while interpreting the results of this study. To guide the reader, this issue is pointed out as one of the limitations of this study. At the same time we still believe that the results of this study are adequate and of value. We chose this original exercise program because we wanted the prevention program to be time efficient (part of warm up, without the need of extra practice), ready to use regardless of the availability of training equipment, medical personnel or coaches. We agree that since no effect of this exercise-based injury-prevention program on injury risk, motor per-formance or psychosocial status could be detected future interventions should use different exercise program.